# Preserved Executive Control in Ageing: The Role of Literacy Experience

**DOI:** 10.3390/brainsci12101392

**Published:** 2022-10-14

**Authors:** Ana I. Pérez, Georgia Fotiadou, Ianthi Tsimpli

**Affiliations:** 1Theoretical and Applied Linguistics, University of Cambridge, Cambridge CB3 9DA, UK; 2Department of Experimental Psychology, University of Granada, 18011 Granada, Spain; 3Department of Linguistics, Aristotle University of Thessaloniki, 54124 Thessaloniki, Greece

**Keywords:** healthy ageing, cognitive decline, executive control, literacy experience, cognitive reserve

## Abstract

Healthy ageing is commonly accompanied by cognitive decline affecting several domains such as executive control, whereas certain verbal skills remain relatively preserved. Interestingly, recent scientific research has shown that some intellectual activities may be linked to beneficial effects, delaying or even alleviating cognitive decline in the elderly. Thirty young (age: *M* = 23) and thirty old (age: *M* = 66) adults were assessed in executive control (switching) and literacy experience (print exposure). First, we tried to confirm whether healthy ageing was generally associated with deficits in switching by looking at mixing cost effects, to then investigate if individual differences in print exposure explained variation in that age-related mixing costs. Both accuracy and reaction times mixing cost indexes demonstrated larger cost in old (but not in young) adults when switching from local to global information. More importantly, this cost effect was not present in old adults with higher print exposure (reaction times). Our findings suggest literacy experience accumulated across the life-span may act as a cognitive reserve proxy to prevent executive control decline.

## 1. Introduction

In the absence of any neurophysiological or neuropsychological disease, healthy ageing is accompanied by cerebral changes which usually cause a decline in several functions such as motor control, emotion and cognition [1]. Age-related deficits have been linked to a variety of cognitive domains such as episodic memory, working memory, lexical access and learning, implicit learning, as well as visuospatial skills [2,3]. Moreover, extensive literature has narrowed down cognitive decline in healthy ageing with executive control. An age-related loss in inhibitory control has been well documented [4,5], which is especially relevant taking into account the fact that inhibition is the common factor underlying most executive functions [6]. In addition, old adults are typically slower and more error prone than young adults in tasks where switching is required [7]. Specifically, a typical paradigm used to assess switching is the global–local task. In this task, participants are shown a global stimulus (e.g., letter H or S) formed by local stimuli (e.g., smaller letters) that are either the same as (congruent trials) or different from (incongruent trials) the global pattern. Participants are instructed to identify the global or local stimulus in single-task blocks (where only local or global trials are required) and in task-mixed blocks (where both local and global trials alternate). In general, global information is processed faster and more accurately than local information [8], causing interference when local identification is requested within a mixed block.

Furthermore, the literature on task switching distinguishes between two types of costs: (a) *s**witching costs*, which refer to the difference between switch and non-switch trials within task-mixed blocks (also named local or specific switch costs), and (b) *mixing costs*, that is, the difference between trials in task-mixed blocks and trials in single-task blocks (also named global or general switch costs; [7,9]). More importantly, while most studies investigating age-related deficits with task switching do not find any (or only small) switching costs associated with healthy ageing, nearly all of them manifest increased mixing costs in old compared to young adults [10], and in particular when mixing cost occurs in the global stimulus [9,11]. This age-related effect associated with mixing cost has been interpreted as reflecting old adults’ difficulty to select and maintain task sets over time and/or update internal control settings imposed by the task [9,10,12]. Taking into account the relevance of mixing (but no of switching) cost in healthy ageing executive decline, from now on we will focus only on mixing cost.

While age-related cognitive decline has been well documented, very promising scientific research has also found that some cognitive functions are fully preserved during healthy ageing. For instance, beneficial age-related cognitive effects have been associated with (crystallized) intellectual activities such as practicing music [13], learning a complex skill such as digital photography [14], achieving a demanding occupational attainment [15], or acquiring a second language [16]. More importantly, verbal abilities are usually well preserved up to a very advanced age. In fact, although processing speed may be somewhat slower than in young adults, old adults seem to perform equally well in linguistic tasks requiring lexical and morphosyntactic processing [17,18], and even better than young adults in abilities such as vocabulary, lexical–semantic knowledge, verbal comprehension and discourse processing [3,19]. Moreover, although a reduced working memory capacity seems to impair old adults’ comprehension of syntactically complex texts [20], older adults make use of the context to interpret information demonstrating good comprehension of what they are reading (see [21] for a review). 

Crucially, high literacy experience seems to protect against memory decline among healthy ageing [22]. A relevant study to the present research was carried out by Payne and colleagues [23]. These authors evaluated the performance of older adults’ speed and recall on sentence comprehension, as well as assessing print exposure through the author recognition test [24], which combines both a subjective (e.g., self-reported time) and an objective (e.g., observed reading in natural contexts) measure of everyday life reading and literacy activities. They witnessed a relationship between higher print exposure and some comprehension measures (e.g., greater attention allocation to clause wrap-up), suggesting literacy experience may prevent cognitive decline in reading comprehension. Interestingly, they also found that higher print exposure buffered the adverse effects of working memory in old adults, by reducing difficulty in sentence wrap-up. In light of these results, print exposure seems to be a good proxy for investigating preserved executive control in healthy ageing.

### The Present Study

Taking into account the evidence on cognitive and, more specifically, executive deficits in healthy ageing, as well as the attested benefits of higher print exposure on preserved sentence comprehension in the elderly, the aim of the present study is two-fold. First, we seek to confirm the relationship between healthy ageing and cognitive control decline. To do this, we use a global–local task to assess young and old adults’ mixing cost effects (accuracy and reaction times). We predict that old adults will present larger mixing costs than young adults, indicating age-related decline in executive control. Second, because it is still unknown whether literacy experience accumulated across the life-span may help to reduce executive control decline, we investigate whether individual differences in print exposure explain age-differences in mixing cost effects. To this end, we evaluate print exposure by means of two independent tests (see Section 2.2.2) and include their composite score as a continuous measure in the statistical model. Higher print exposure is expected to be associated with a reduction in mixing cost effects specifically in old adults, which would indicate that literacy experience may compensate for executive control decline. 

## 2. Method

### 2.1. Participants

Participants were 60 native Greek speakers: 30 young adults (age: *M* = 23.37, *SD* = 3.88; range = 18–32; 18 women) and 30 old adults (age: *M* = 66.27, *SD* = 5.49; range = 59–78; 17 women). All participants gave their informed consent prior to testing and volunteered to participate.

### 2.2. Materials

Our materials are divided into two sections: (a) control measures and (b) experimental tasks. First, several measures were used to dismiss the possibility of cognitive impairment in the old group, as well as to control for general group differences in educational background, fluid intelligence and working memory. Second, the global–local task was assessed to extract the mixing cost indices regarding both accuracy and reaction times. Finally, print exposure was assessed to understand whether mixing cost effects were explained by individual differences in literacy experience. 

#### 2.2.1. Control Measures

Cognitive impairment. The Mini-Mental State Examination (MMSE, [25]) was used in its Greek adaptation [26] to discard any possible cognitive impairment (e.g., dementia) in old adults. Participants’ performance was almost at ceiling (*M* = 29.09, *SD* = 1.04, range = 26–30), suggesting no cognitive deterioration in the elderly.

Educational background. A short questionnaire on demographic information was administered to control for group differences in the amount of years of formal education. A *t*-test comparison on this measure manifested no significant effect of group, *t*(58) = 1.67, *p* = 0.10, where young (*M* = 14.40, *SD* = 1.02) and old (*M* = 15.47, *SD* = 3.31) adults had similar educational backgrounds.

Fluid intelligence. The Raven’s Advanced Progressive Matrices test [27], was used as a standardized nonverbal individual test measuring fluid intelligence (abstract reasoning). Participants were presented with 60 matrices divided into five sets of 12 patterns assessing visuospatial abilities (set A and the first half of set B assessed), and analytic abilities such as problem solving (second half of set B, set C, D and E; [28]). A *t*-test comparison on this task showed a significant effect of group, *t*(58) = −2.63, *p* = 0.01, where young adults (*M* = 46.67, *SD* = 8.24) performed significantly better than old adults (*M* = 40.34, *SD* = 10.26). Despite the fact that this difference between young and old adults in fluid intelligence was an unwanted effect, we believe it cannot explain any possible finding exclusively found within the old group (see Discussion). 

Working Memory. Working memory capacity was assessed by the backward digits span task of the Automated Working Memory Assessment (AWMA, [29]). List of digits were auditorily presented, and participants had to recall the sequence of digits in the reverse order. There were six blocks increasing in difficulty from two to seven digits, with five trials per block. Participants continued with the next block if they recalled four out of six trials. In contrast, when participants failed at least three trials of the same block, the task finished. The final score was the total number of trials correctly recalled. A *t*-test comparison on working memory scores demonstrated no significant effect of group, *t*(58) = −1.00, *p* = 0.32, where young (*M* = 19.50, *SD* = 5.14) and old adults (*M* = 18.07, *SD* = 5.97) had similar working memory capacity.

#### 2.2.2. Experimental Tasks

Global–local task. The global–local task was administered (by the E-prime software, [30]) to assess switching. Stimuli appeared in the center of the screen until participants provided a response. We used shapes instead of letters to increase the familiarity of stimuli [31]. Concretely, the stimuli were circles, triangles, squares, or Xs for neutral trials [32,33]. For example, if a large square composed of small triangles appeared, participants had to respond according to the large figure by indicating “square” (global condition), or according to the small figures by signaling “triangle” (local condition). There were 3 experimental blocks with 64 trials each: one global and one local (single-task blocks), and one combining both global and local trials (task-mixed block). Instructions preceding each block informed participants whether the target was the global or local condition (in single-task blocks) or whether participants had to switch after each trial, alternating between the global and local (in task-mixed block). In addition, if participants made five mistakes in a row in the task-mixed block, the global/local condition was reminded by indicating the next correct response (less than 5% for old adults and no cases for young adults). Each block was composed of: (a) congruent trials, which contained the same shape in the large and small figures; (b) incongruent trials, with different shapes in the large and small figures; and (c) neutral trials, with Xs either in the large or the small figures, requiring just one response option (the figure not containing X/s). Participants provided their responses by pressing one out of four keys (in relation to the circle, triangle, square or X shape), counterbalanced across participants. Both accuracy and reaction times were measured. A total of 5 practice trials were provided for each experimental block to ensure participants understood the task, and breaks were allowed between blocks to prevent fatigue. The task took about 10–15 min. 

The mean of both non-switching trials (single-task blocks) and switching trials (task-mixed block) were extracted to calculate two *mixing cost indices*: *accuracy* (proportion), which was calculated by subtracting the non-switching trials to switching trials, and *reaction times* (*RTs* in milliseconds), which were computed only for correct responses by using the reverse pattern, that is, subtracting the switching trials to non-switching trials.

Author Recognition Test (ART). An adapted version of the Author Recognition test [24] was developed in Greek [34]. In this test, participants were presented with a checklist of 80 items: half were names of literature writers (40 targets) and half names of authors from other genres (40 foils). Participants had to indicate which items consisted of names of literature authors. The final score was obtained by subtracting the total number of correct target responses from the total number of false foil responses.

Magazine Recognition Test (MRT). Similarly, an adapted version of the Magazine Recognition test [24] was developed in Greek [34]. Here, participants were presented with a checklist of 80 items: half were names of well-known magazines (40 targets) and half names of newspapers, TV shows, series and movies displayed in the Greek TV program at the time of the study (40 foils). Participants had to indicate which items were names of real magazines. Once more, scores were calculated by subtracting the number of target responses from the number of foil responses.

Print exposure scores were calculated by adding up the extracted ART and MRT z-scores per participant, as a measure of *literacy experience*. 

### 2.3. Procedure

There were three experimental sessions of approx. 30–40 min., administered in a timeframe of 7–9 days. In all of them, participants were assessed in a quiet room. The first session began with the demographic’s questionnaire, followed by the MMSE to assess cognitive impairment. The second session started with the global–local task, then the working memory task, and ended with the print exposure tests. Finally, in the third session, participants performed the fluid intelligence task.

### 2.4. Data Analysis

Our analyses were carried out with 60 participants (30 per group) and 192 trials (96 global and 96 local trials; 32 trials per crossed condition) in total. Linear mixed-effects (LME) models were performed on the two mixing cost indices of accuracy and RTs, using the *lmer* function of the lme4 R package. Each LME model contained Participants as the random factor (the random factor of Items was not included in the analyses Because the two mixing cost indices were computed by averaging items in each crossed condition), and Group (young vs. old), Condition (global vs. local), Congruence (congruent vs. incongruent vs. neutral), and Print exposure (continuous factor), as the fixed factors. RTs values going lower than 200 ms. or higher than 5000 ms. (1.78%) were eliminated. In addition, extreme outlier RTs data per group, condition and congruence were cleaned (2.23%), before calculating the mixing cost index of RTs. The *simr* package was used to run power analyses with 1000 simulations and alpha set at 0.05 in each LME model (accuracy and RTs), with the purpose of assessing whether our sample size was sufficient to defend the significant effects that were found. Results of these power analyses confirmed that the sample size was sufficient for each model: Accuracy = 100% (CI = 99.63–100%) and RTs = 99.80% (CI = 99.28–99.98%) [35]. 

Regarding LME models, first, keeping the full fixed structure, we looked for the best random structure, which was the same for both accuracy and RTs (condition |participants). Second, the best fixed structure was found using stepwise model comparison from the most complex model to the simplest, and selecting the one with lower AIC and BIC, and significant *χ^2^* test for the Log-likelihood, using the maximum likelihood (for details on this procedure see [36]). The *p*-value of each significant fixed effect was calculated by the *anova* function of the lmerTest R package. Moreover, post-hoc comparisons with Bonferroni correction were conducted by the *testInteractions* function of the phia R package.

## 3. Results

Our results are organized into two sections. First, we report the mixing cost effects found in accuracy, and subsequently the same effects in RTs (see Table 1 for means and standard deviations in single-task blocks, task-mixed blocks and mixing cost). Each LME model was composed of a four-way interaction including Group (young vs. old adults), Condition (global vs. local), Congruence (congruent vs. incongruent vs. neutral) and Print exposure. Taking into account the large number of results presented in this study, we focused on the fixed effects of each LME model. Summary details (lmerTest package) regarding models fit and random effects are provided in the Appendix.

### 3.1. Mixing Cost in Accuracy

The four-way interaction on accuracy demonstrated the significant main effects of group, *F* (1, 63) = 11.52, *p* < 0.01, *η_p_^2^* = 0.16, where old adults presented larger mixing cost (lower accuracy) than young adults; condition, *F* (1, 62) = 20.08, *p* < 0.001, *η_p_^2^* = 0.25, with larger mixing cost in the global compared to the local condition; and congruence, *F* (2, 248) = 22.08, *p* < 0.001, *η_p_^2^* = 0.15, where both the incongruent and neutral trials caused larger cost than the congruent trials. In addition, four two-way interactions were significant: group and condition, *F* (1, 61) = 4.76, *p* < 0.05, *η_p_^2^* = 0.07, group and congruence, *F* (2, 248) = 4.13, *p* < 0.05, *η_p_^2^* = 0.03; condition and congruence, *F* (2, 248) = 7.05, *p* < 0.01, *η_p_^2^* = 0.05; and condition and print exposure, *F* (1, 61) = 7.51, *p* < 0.01, *η_p_^2^* = 0.11. No other effects reached significance (all *ps* > 0.05). To understand the nature of these interactions, we ran post-hoc comparisons with Bonferroni correction. 

On the one hand, the interaction of group and condition showed larger cost in the global compared to the local condition in old adults, *χ^2^* (1) = 21.25, *p* < 0.001, but not in young adults, *χ^2^* (1) = 1.83, *p* = 0.35 (see Figure 1a). That is, regardless of the type of trial, switching from local to global information was especially demanding for old participants. On the other hand, the interaction of group and congruence manifested a strong difference between trials in old adults, *χ^2^* (2) = 47.24, *p* < 0.001, but only a tendency to significance in young adults, *χ^2^* (2) = 6.87, *p* = 0.06. Specifically, compared to the congruent trials, old adults presented larger cost in the incongruent, *t* (130) = 5.33, *p* < 0.001, and neutral trials, *t* (130) = 4.37, *p* < 0.001, with no differences between these two, *t* (129) = 0.98, *p* = 0.33 (see Figure 1b). This finding suggested that difficulties imposed by the conflicting (incongruent) and the apparently non-conflicting (neutral) situations were particularly challenging, especially for old adults. 

The interaction of condition and congruence also showed larger cost changing from the local to the global condition in the incongruent, *χ^2^* (1) = 21.08, *p* < 0.001, and neutral trials, *χ^2^* (1) = 16.72, *p* < 0.001, but not in the congruent trials, *χ^2^* (1) = 0.17, *p* ≃ 1.00. This effect signaled difficulty switching from local to global information when conflict was imposed (incongruent), but also, when non-conflict was presented (neutral). 

More importantly, the interaction between condition and print exposure indicated that lower print exposure was associated with larger cost in the global, *χ^2^* (1) = 6.94, *p* < 0.05, but not in the local condition, *χ^2^* (1) = 0.01, *p* ≃ 1.00. In contrast, higher print exposure reduced this cost effect (see Figure 2). 

### 3.2. Mixing Cost in RTs

The four-way interaction on RTs showed significant two-way interactions of group and condition, *F* (1, 62) = 13.10, *p* < 0.001, *η_p_^2^* = 0.17, and condition and congruence, *F* (2, 240) = 5.01, *p* < 0.01, *η_p_^2^* = 0.04. More importantly, the three-way interaction of group, condition and print exposure was also significant, *F* (1, 61) = 7.48, *p* < 0.01, *η_p_^2^* = 0.11. No other effect reached significance (all *ps* > 0.05). Once more, post-hoc comparisons with Bonferroni correction were run in each interaction.

Similar to the accuracy findings, the interaction of group and condition demonstrated larger mixing cost (longer RTs) in the global compared to the local condition in old adults, *χ^2^* (1) = 9.90, *p* < 0.01, but not in young adults, *χ^2^* (1) = 0.01, *p* ≃ 1.00 (see Figure 3). Thus, once again, regardless of the type of trial, switching from local to global information was more difficult, particularly for old adults. 

The interaction of condition and congruence manifested larger mixing cost when changing from the local to the global condition this time only in the neutral trials, *χ^2^* (1) = 13.56, *p* < 0.001, but not in either the congruent, *χ^2^* (1) = 0.01, *p* ≃ 1.00, or incongruent trials, *χ^2^* (1) = 0.59, *p* ≃ 1.00. Thus, once more, the apparently non-conflict (neutral) condition caused difficulty when switching from local to global information. 

Finally, to follow up on the three-way interaction, we divided the analysis by group. The remaining condition and print exposure interaction was significant in old adults, *F* (1, 32) = 7.45, *p* < 0.05, but not in young adults, *F* (1, 30) = 2.18, *p* = 0.15. This interaction came from opposite regression slopes between the local and the global conditions, *χ^2^* (1) = 7.45, *p < 0*.01, where lower print exposure was related to larger cost in the global compared to the local condition (see Figure 4). These results suggest that previous observed difficulties to switch from local to global information were specifically affecting old adults with lower (but not higher) print exposure.

## 4. Discussion

The aim of the present study was two-fold. On the one hand, we tried to confirm previous evidence showing a relationship between healthy ageing and executive control decline, by demonstrating larger mixing cost in old compared to young adults. On the other hand, we investigated whether this possible effect was buffered by higher print exposure exclusively in old adults, indicating that literacy experience might compensate for executive control healthy decline. Following these aims, we first discuss general effects found with the global–local paradigm to then focus on our two research goals.

### 4.1. General Global–Local Effects

The model on accuracy showed that, in general, participants experienced difficulty in switching from the local to the global condition. This mixing cost effect from local to global information can be easily interpreted by the global advantage or superiority effect. That is, global processing is usually faster and more accurate than local processing and thus, when the latter is recruited, it is assumed to cause interference in subsequent global information [8]. Moreover, both mixing cost indexes showed that this effect was particularly evident when information was either incongruent (accuracy) or neutral (accuracy and RTs). From our point of view, these results reflect two different findings. First, the incongruent effect demonstrates that it was especially difficult to switch from local to global information when there was a conflict between the global and local stimuli. This is consistent with the vast majority of studies using the global–local task [37,38] and it suggests that when circumstances impose a highly demanding ambiguous situation (incongruent), cognitive sources are reduced affecting the other task’s demands (global vs. local conditions). 

Second, taking into account that neutral trials (which only required one response) were supposed to impose no conflict, the fact that we found accuracy and RT effects with neutral trials was somehow unexpected. We believe this could be due to the nature of our stimuli. Dukette and Stiles [31] suggested that shapes are more familiar than letters, and therefore we used shapes (circles, triangles or squares) for the congruent and incongruent trials to make it easier for older participants. However, the neutral trials (Xs) can be interpreted as both a shape and/or a letter. According to Bialystok [32], this ambiguity in the neutral trials attracts attention to the opposite level (global or local), generating interference. Indeed, the same effect was previously found in both [32]’s and [31]’s studies. In relation to this, our findings suggest participants generally experienced some level of interference when responding to ambiguous information. Nonetheless, further research would be necessary to clarify this issue.

### 4.2. Age-Related Executive Decline

As expected, old adults had larger mixing cost (lower accuracy) than young adults. Moreover, in line with previous research, these age-related differences were related to a global–local effect. Both accuracy and RTs indices showed larger cost in old adults when they switched from local to global information, whereas the same was not true for young adults. This finding suggests that old adults experienced problems changing the task set when attention was previously guided towards local information, replicating previous results found with the global–local task, where old adults were slower in the global condition when RTs from single-task blocks were compared to task-mixed blocks [9]. 

Similar to prior results, the cost to switch from local to global information can be explained by the global superiority effect. However, in this case, mixing costs were proven to be true only in old adults, signaling this population had difficulties implementing executive control. Although the literature on task switching has not yet identified the specific executive functions that constrain age-related differences in this regard, mixing cost effects in old adults have been suggested to reflect problems with the control processes that are necessary for selecting and maintaining several task sets in a switching situation [10,12]. Notice however, that a maintenance deficit interpretation would be less probable in the present study, as working memory capacity was similar across age-groups. In addition, other researchers have argued that these problems are related to difficulties in a set-updating process that “cleans up” the internal control settings when the task context causes ambiguity [9]. Furthermore, old adults had larger cost in the incongruent and neutral trials compared to the congruent trials (accuracy), indicating that the conflicting/ambiguous situation created by these two types of trials was especially demanding for the elderly.

Overall, our findings demonstrate deficits in executive control in old healthy adults, which seem to be associated with the executive functions of switching, updating and inhibition (see [6,39]). A crucial empirical question here is: Could individual differences in print exposure buffer this executive control decline? In the following section we tried to address this question.

### 4.3. Literacy Experience and Cognitive Reserve

Everyday life intellectual activities may help to preserve cognitive functioning during ageing [13,14,15,16,22]. Interestingly, old adults perform equally well or even better than young adults in verbal skills such as vocabulary, verbal comprehension and discourse processing [3,19]. In fact, they make use of the context to interpret information, demonstrating good reading comprehension [21]. Accordingly, in the present study we further investigated if the age-related executive control decline found in the global–local task was diminished by a plausible cognitive reserve proxy: Literacy experience.

Both accuracy and RTs mixing cost indices manifested a relationship between lower print exposure and larger cost in the global compared to the local condition, suggesting print exposure was a general predictor for the global–local effect. Interestingly, for RTs, this effect was qualified by a three-way interaction with group, which indicated that the negative relationship between mixing cost and literacy experience was only true for old adults. As predicted, lower print exposure was associated with larger cost in the global compared to the local condition in old adults, whereas this effect was absent in young adults. More importantly, old adults with higher print exposure showed a decrease in the global–local differences, indicating that they were better at exerting more efficient executive control when switching from local to global information. These results are in line with [23] findings, where higher print exposure (also assessed by ART [24]) was related to better comprehension skills (such as greater attention allocation to clause wrap-up) in healthy ageing and to a compensation of working memory decline by facilitating wrap-up effects and sentence recall. Our findings go beyond the beneficial impact of print exposure on verbal materials to transfer it into a non-verbal (non-domain specific) executive control advantage. 

The beneficial long-lasting effect of print exposure in executive control can be understood if we conceptualize literacy as a crystallized intellectual ability. In fact, print exposure represents the accumulated reading experience a person has acquired throughout his/her lifetime, a dimension that is directly linked to our prior knowledge, and therefore, to our crystalized intelligence [40]. Recent neuroscientific evidence has shown that print exposure is correlated with grey matter cortical thickness in the left inferior frontal gyrus and the left fusiform gyrus, suggesting reading experience might be connected to maturational brain processes that are related to skill consolidation [41,42]. In addition, print exposure has been associated with many verbal abilities such as vocabulary size, syntactic and phonological processing, verbal fluency, and other higher-level text comprehension processes [43], usually related to crystallized rather than fluid intelligence [44,45]. Thus, the connection between literacy experience (as a crystallized intellectual activity) and executive functions seems to be apparent. However, how can literacy experience preserve our executive control processes?

Literacy might be considered a general domain (rather than domain-specific) ability by which a person trains multiple executive functions [46]. For instance, text comprehension usually requires: (1) the activation of a vast amount of information in working memory, (2) metacognitive skills to detect inconsistencies that may appear as a product of misunderstandings, (3) the inhibition of no longer relevant information, and (4) constant updating of text information in order to build a coherent mental representation, among other processes [36]. Furthermore, literacy experience could enhance basic cognitive processes (short-term and long-term memory skills), that would translate into faster speed via associative processing [40]. Indeed, higher print exposure frequently reflects faster reading times and therefore, faster processing speed. Accordingly, it seems reasonable to think that old adults with higher print exposure performed more efficient executive control processes by showing faster RTs in correct responses. 

Finally, an important limitation of our study is the fact that the two groups differed in fluid intelligence, with a lower performance in old compared to young adults. Although this difference cannot account for the most relevant finding of this article (i.e., the relationship between individual differences in print exposure and executive control exclusively found in the old group), it could have impacted on other effects manifesting group differences. Accordingly, further research would be necessary to clarify this issue.

## 5. Conclusions

To sum up, the present study extends previous research into executive control decline in healthy ageing and compensatory mechanisms by showing that literacy experience may be a good cognitive reserve proxy for executive control. Using a global–local task, we found that both accuracy and RTs mixing cost indices reflect (1) difficulty in old compared to young adults when switching from local to global information; and (2) a relationship between higher print exposure (RTs) and a reduction of the global–local interference effect, specifically in old adults. Therefore, our study suggests that literacy experience may preserve executive functions during healthy ageing.

## Figures and Tables

**Figure 1 brainsci-12-01392-f001:**
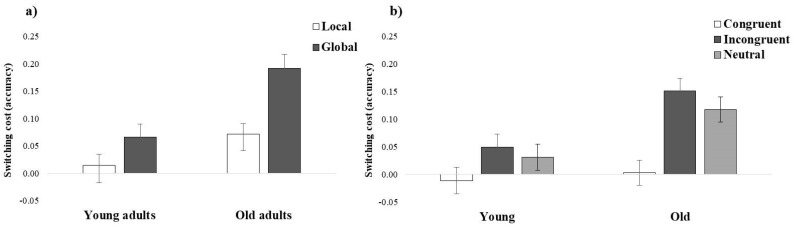
Mixing cost in accuracy (proportion, i.e., non-switching trials minus switching trials) obtained in the global-local task as a function of (**a**) group and condition, and (**b**) group and congruence.

**Figure 2 brainsci-12-01392-f002:**
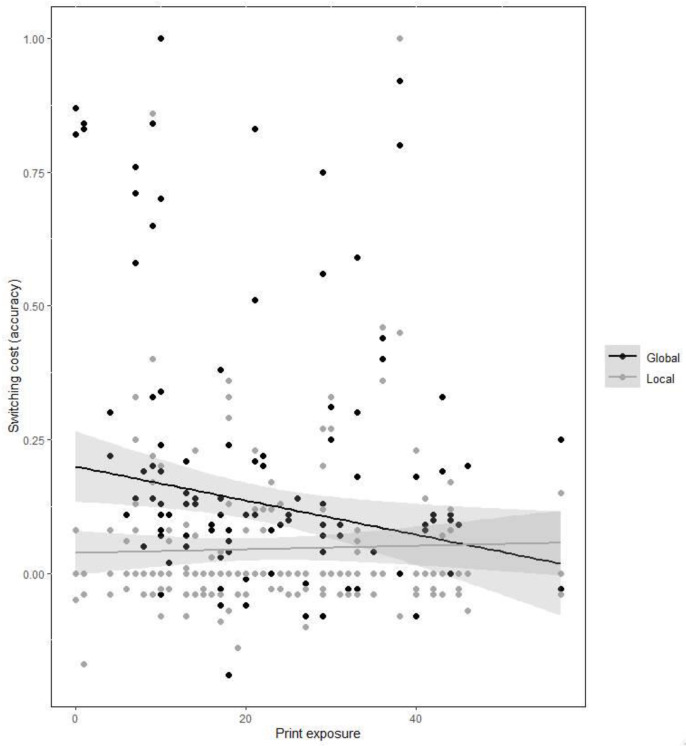
Mixing cost in accuracy (proportion, i.e., non-switching trials minus switching trials) obtained in the global-local task as a function of condition and print exposure.

**Figure 3 brainsci-12-01392-f003:**
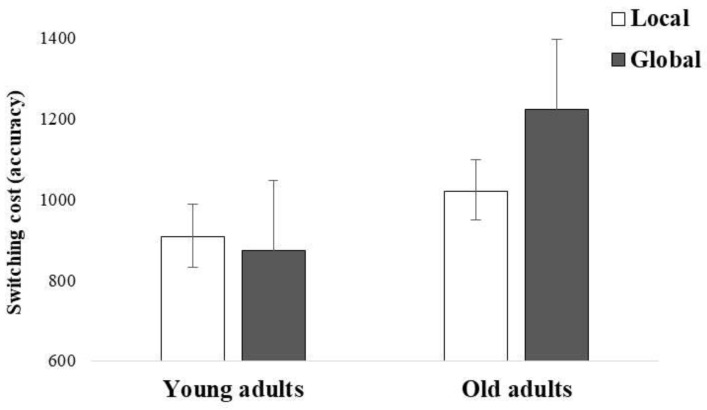
Mixing cost in reaction times (in milliseconds, i.e., switching trials minus non-switching trials) obtained in the global-local task as a function of group and condition.

**Figure 4 brainsci-12-01392-f004:**
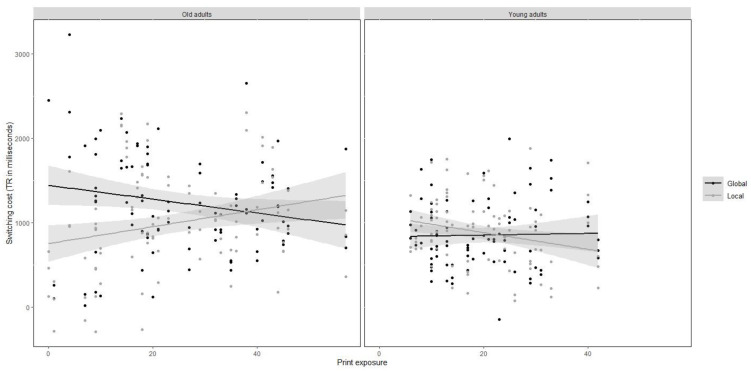
Mixing cost in reaction times (in milliseconds, i.e., switching trials minus non-switching trials) obtained in the global-local task as a function of group, condition and print exposure (literacy experience).

**Table 1 brainsci-12-01392-t001:** Means and standard deviations for single-task blocks, task-mixed blocks, and mixing cost in accuracy (proportion) and reaction times (in milliseconds) on the global–local task, as a function of group, condition and congruence.

			Single-Task Blocks		Task-Mixed Blocks		Mixing Cost	
			Local	Global	Local	Global	Local	Global
**Accuracy**	Young	Congruent	1.00 (0.02)	1.00 (0.02)	0.99 (0.04)	0.99 (0.03)	0.01 (0.04)	0.01 (0.04)
		Incongruent	0.97 (0.03)	0.96 (0.03)	0.95 (0.09)	0.86 (0.14)	0.02 (0.09)	0.11 (0.13)
		Neutral	0.98 (0.03)	0.98 (0.03)	0.97 (0.08)	0.90 (0.13)	0.01 (0.07)	0.08 (0.13)
	Old	Congruent	0.99 (0.02)	1.00 (0.02)	0.97 (0.09)	0.95 (0.11)	0.03 (0.10)	0.04 (0.11)
		Incongruent	0.97 (0.02)	0.92 (0.17)	0.86 (0.25)	0.65 (0.35)	0.11 (0.26)	0.27 (0.32)
		Neutral	0.97 (0.05)	0.94 (0.15)	0.89 (0.16)	0.71 (0.33)	0.08 (0.16)	0.24 (0.33)
**RTs**	Young	Congruent	1054 (242)	1036 (175)	1985 (490)	1759 (509)	931 (468)	722 (424)
		Incongruent	1128 (222)	1082 (160)	2134 (419)	1998 (500)	1005 (373)	915 (404)
		Neutral	1157 (242)	1067 (164)	1940 (397)	2024 (445)	783 (354)	957 (351)
	Old	Congruent	1316 (229)	1342 (228)	2361 (630)	2498 (739)	1045 (611)	1165 (720)
		Incongruent	1392 (234)	1412 (229)	2447 (544)	2623 (604)	1049 (544)	1256 (557)
		Neutral	1465 (288)	1370 (230)	2406 (671)	2630 (678)	942 (659)	1260 (561)

Note. Mixing cost for accuracy was calculated by subtracting the non-switching trials to switching trials, whereas for RTs was done by subtracting the switching trials to non-switching trials only for correct responses.

## Data Availability

All data for the study reported here are available at www.osf.io using the following link: https://osf.io/bfkxn/?view_only=324e092e99184fb585af238671f290bf accessed on 30 March 2022). In addition, a preprint of this manuscript has been submitted to bioRxiv.

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
