# Peer review of "Preserved Executive Control in Ageing: The Role of Literacy Experience"

_brainsci, 2022, doi:10.3390/brainsci12101392_

Round 1

Reviewer 1 Report

Thank you very much for the opportunity to review this manuscript. The manuscript is clear, concise and to the point, and substantially contributes to current knowledge on the topic. I have, however, some concerns that hopefully can be addressed by the Authors.

First of all, the participant sample is quite limited, and given that healthy ageing adults are considered, the sample could be relatively easily expanded. I would recommend expanding the sample to improve the generalisability of the results. If this is not possible (and it typically is not at this point of a study), please include a power calculation to strengthen the choice of this sample size.

Second, I would recommend discussing limitations of this research in a separate sections. In the current version of the manuscript, limitations are either not stated or are successfully hidden in the discussion.

Third, in the Results section, I recommend clearly showing the implemented models and their results. As the results stand now, it is difficult to understand whether four-way or three-way interactions were found and how precisely Authors dealt with them. Tables with models and their parameters would likely help. Please also clarify the first paragraph in section 3.1. Is the paragraph opened with a four-way interaction? If so, why are main effects reported - they are not meaningful if the variables are implicated in interactions.

Minor comments:

line 20. Do Authors mean "was not present", not "disappeared"?

line 63. "On the other hand" suggests an upcoming counter-argument. Since what follows does not counter what was said before (only expands it), I would suggest using another phrase, e.g., "While [xxx] decline, very promising..."

line 182. I would recommend spelling out "the Author Recognition Test" and "the Magazine Recognition Test" separately.

line. 301. "was" - do the Authors mean "were"?

Author Response

First of all, the participant sample is quite limited, and given that healthy ageing adults are considered, the sample could be relatively easily expanded. I would recommend expanding the sample to improve the generalisability of the results. If this is not possible (and it typically is not at this point of a study), please include a power calculation to strengthen the choice of this sample size.

Authors: We thank Reviewer 1 for bringing the issue of the sample size. Unfortunately, at this point, it would be almost impossible to collect more participants to increase the sample, as the main author of this work is no longer in Greece, were the original sample was recruited. However, as suggested by the Reviewer, power analyses have been conducted for each statistical model, and we have found enough statistical power (over 80%) in both of them: “The simr package was used to run power analyses with 1,000 simulations and alpha set at .05 in each LME model (accuracy and RTs), with the purpose of assessing whether our sample size was sufficient to defend the significant effects that were found. Results of these power analyses confirmed that the sample size was sufficient for each model: Accuracy = 100% (CI = 99.63 - 100%) and RTs = 99.80% (CI = 99.28 - 99.98%) [35].” (see page 5).

Second, I would recommend discussing limitations of this research in a separate section. In the current version of the manuscript, limitations are either not stated or are successfully hidden in the discussion.

Authors: We agree with Reviewer 1, as no limitation was explicitly stated. We have now provided a paragraph were the main limitation is discussed just before the Conclusions: “Finally, an important limitation of our study is the fact that the two groups differed in fluid intelligence, with a lower performance in old compared to young adults. Although this difference cannot account for the most relevant finding of this article (i.e., the relationship between individual differences in print exposure and executive control exclusively found in the old group), it could have impacted on other effects manifesting group differences. Accordingly, further research would be necessary to clarify this issue.” (page 11).

Third, in the Results section, I recommend clearly showing the implemented models and their results. As the results stand now, it is difficult to understand whether four-way or three-way interactions were found and how precisely Authors dealt with them. Tables with models and their parameters would likely help. Please also clarify the first paragraph in section 3.1. Is the paragraph opened with a four-way interaction? If so, why are main effects reported - they are not meaningful if the variables are implicated in interactions.

Authors: First, we have clearly stated that a four-way interaction was implemented in both models: “Our results are organised in two sections. First, we report the mixing cost effects found in accuracy, and subsequently the same effects in RTs (see Table 1 for means and standard deviations in single-task blocks, task-mixed blocks and mixing cost). Each LME model was composed of a four-way interaction including Group (young vs. old adults), Condition (global vs. local), Congruence (congruent vs. incongruent vs. neutral) and Print exposure.” (page 5). In addition, this has been also specified at the beginning of each Results section: 1) 3.1. Mixing cost in accuracy: “The four-way interaction on accuracy demonstrated…” (page 6) and 2) 3.2. Mixing cost in RTs: “The four-way interaction on RTs showed…” (page 7).

Second, we have followed a systematic procedure in the Results section by which all significant effects have been reported, starting from the lower level (main effects) and following with higher order effects (first two-way interactions, then three-way interactions, etc.). We agree with Reviewer 1 on the fact that when higher-order effects are significant, main effects may be less relevant. However, it is still informative to know if they were significant or not, and this is why we usually report them (see e.g., Pérez, Hansen & Bajo, 2019; Pérez, Joseph, Bajo & Nation, 2016; Pérez, Schmidt, Kourtzi & Tsimpli, 2020). More importantly, four two-way interactions (accuracy model) as well as two two-way interactions and a three-way interaction (RTs model) were also reported.

Moreover, we informed that the significant two-way interactions were analysed by running post-hoc comparisons with Bonferroni correction. Nonetheless, Reviewer 1 is right stating that the procedure was not completely explained. To solve this, we have included the following information in the Data analysis section: “The p-value of each significant fixed effect was calculated by the anova function of the lmerTest R package. Moreover, post-hoc comparisons with Bonferroni correction were conducted by the testInteractions function of the phia R package.” (page 5).

Finally, for the three-way interaction found in the RTs model, it was explained that: “…to follow up on the three-way interaction, we divided the analysis by group.” (page 8). This division by group means that we ran post-hoc comparisons with Bonferroni correction in the two-way interaction of condition and print exposure for young and old adults separately, as it is implicitly reflected in the following sentence: “The remaining condition and print exposure interaction was significant in old adults, F (1, 32) = 7.45, p < .05, but not in young adults, F (1, 30) = 2.18, p = .15.” (page 8).

Minor comments:

line 20. Do Authors mean "was not present", not "disappeared"?

Authors: Changed (see Abstract).

line 63. "On the other hand" suggests an upcoming counter-argument. Since what follows does not counter what was said before (only expands it), I would suggest using another phrase, e.g., "While [xxx] decline, very promising..."

Authors: Changed to: “While age-related cognitive decline has been well documented, very promising scientific research has found that some cognitive functions are fully preserved during healthy ageing.” (see page 2)

line 182. I would recommend spelling out "the Author Recognition Test" and "the Magazine Recognition Test" separately.

Authors: The two tasks have been spelled out (see pages 4-5):

“Author Recognition Test (ART). An adapted version of the Author Recognition test [24] was developed in Greek [34]. In this test, participants were presented with a checklist of 80 items: half were names of literature writers (40 targets) and half names of authors from other genres (40 foils). Participants had to indicate which items consisted of names of literature authors. The final score was obtained by subtracting the total number of correct target responses from the total number of false foil responses.

Magazine Recognition Test (MRT). Similarly, an adapted version of the Magazine Recognition test [24] was developed in Greek [34]. Here, participants were presented with a checklist of 80 items: half were names of well-known magazines (40 targets) and half names of newspapers, TV shows, series and movies displayed in the Greek TV program at the time of the study (40 foils). Participants had to indicate which items were names of real magazines. Once more, scores were calculated by subtracting the number of target responses from the number of foil responses.

Print exposure scores were calculated by adding up the extracted ART and MRT z-scores per participant, as a measure of literacy experience.”

line. 301. "was" - do the Authors mean "were"?

Authors: Changed (see page 8).

Reviewer 2 Report

the manuscript is highly relevant and interesting. It adds clearly to the subject area compared with other published material.
It is very well written and hence clear and easy to read. The conclusions are consistent with the evidence and arguments presented.

Author Response

REVIEWER 2

The manuscript is highly relevant and interesting. It adds clearly to the subject area compared with other published material.

It is very well written and hence clear and easy to read. The conclusions are consistent with the evidence and arguments presented.

Authors: We thank Reviewer 2 for these kind comments.
